# PAC-Bayesian Theory Meets Bayesian Inference

**Pascal Germain**[†]    **Francis Bach**[†]    **Alexandre Lacoste**[‡]    **Simon Lacoste-Julien**[†]
[†] INRIA Paris - École Normale Supérieure, `firstname.lastname@inria.fr`
[‡] Google, `allac@google.com`

## Abstract

We exhibit a strong link between frequentist PAC-Bayesian risk bounds and the Bayesian marginal likelihood. That is, for the negative log-likelihood loss function, we show that the minimization of PAC-Bayesian generalization risk bounds maximizes the Bayesian marginal likelihood. This provides an alternative explanation to the Bayesian Occam's razor criteria, under the assumption that the data is generated by an *i.i.d.* distribution. Moreover, as the negative log-likelihood is an unbounded loss function, we motivate and propose a PAC-Bayesian theorem tailored for the sub-gamma loss family, and we show that our approach is sound on classical Bayesian linear regression tasks.

## 1   Introduction

Since its early beginning [24, 34], the PAC-Bayesian theory claims to provide "PAC guarantees to *Bayesian* algorithms" (McAllester [24]). However, despite the amount of work dedicated to this statistical learning theory—many authors improved the initial results [8, 21, 25, 30, 35] and/or generalized them for various machine learning setups [4, 12, 15, 20, 28, 31, 32, 33]—it is mostly used as a *frequentist* method. That is, under the assumptions that the learning samples are *i.i.d.*-generated by a data-distribution, this theory expresses *probably approximately correct* (PAC) bounds on the generalization risk. In other words, with probability $1-\delta$, the generalization risk is at most $\varepsilon$ away from the training risk. The *Bayesian* side of PAC-Bayes comes mostly from the fact that these bounds are expressed on the averaging/aggregation/ensemble of multiple predictors (weighted by a *posterior* distribution) and incorporate prior knowledge. Although it is still sometimes referred as a theory that bridges the Bayesian and frequentist approach [*e.g.*, 16], it has been merely used to justify Bayesian methods until now.[1]

In this work, we provide a direct connection between Bayesian inference techniques [summarized by 5, 13] and PAC-Bayesian risk bounds in a general setup. Our study is based on a simple but insightful connection between the Bayesian marginal likelihood and PAC-Bayesian bounds (previously mentioned by Grünwald [14]) obtained by considering the negative log-likelihood loss function (Section 3). By doing so, we provide an alternative explanation for the Bayesian Occam's razor criteria [18, 22] in the context of model selection, expressed as the complexity-accuracy trade-off appearing in most PAC-Bayesian results. In Section 4, we extend PAC-Bayes theorems to regression problems with unbounded loss, adapted to the negative log-likelihood loss function. Finally, we study the Bayesian model selection from a PAC-Bayesian perspective (Section 5), and illustrate our finding on classical Bayesian regression tasks (Section 6).

## 2   PAC-Bayesian Theory

We denote the learning sample $(X, Y) = \{(x_i, y_i)\}_{i=1}^n \in (\mathcal{X} \times \mathcal{Y})^n$, that contains $n$ input-output pairs. The main assumption of frequentist learning theories—including PAC-Bayes—is that $(X, Y)$ is

randomly sampled from a data generating distribution that we denote $\mathcal{D}$. Thus, we denote $(X, Y) \sim \mathcal{D}^n$ the *i.i.d.* observation of $n$ elements. From a frequentist perspective, we consider in this work loss functions $\ell : \mathcal{F} \times \mathcal{X} \times \mathcal{Y} \to \mathbb{R}$, where $\mathcal{F}$ is a (discrete or continuous) set of predictors $f : \mathcal{X} \to \mathcal{Y}$, and we write the empirical risk on the sample $(X, Y)$ and the generalization error on distribution $\mathcal{D}$ as

$$\widehat{\mathcal{L}}_{X,Y}^{\ell}(f) \;=\; \frac{1}{n} \sum_{i=1}^{n} \ell(f, x_i, y_i) \,; \quad \mathcal{L}_{\mathcal{D}}^{\ell}(f) \;=\; \mathop{\mathbf{E}}_{(x,y) \sim \mathcal{D}} \ell(f, x, y) \,.$$

The PAC-Bayesian theory [24, 25] studies an averaging of the above losses according to a *posterior* distribution $\hat{\rho}$ over $\mathcal{F}$. That is, it provides *probably approximately correct* generalization bounds on the (unknown) quantity $\mathbf{E}_{f \sim \hat{\rho}} \mathcal{L}_{\mathcal{D}}^{\ell}(f) \;=\; \mathbf{E}_{f \sim \hat{\rho}} \mathbf{E}_{(x,y) \sim \mathcal{D}} \ell(f, x, y)$, given the empirical estimate $\mathbf{E}_{f \sim \hat{\rho}} \widehat{\mathcal{L}}_{X,Y}^{\ell}(f)$ and some other parameters. Among these, most PAC-Bayesian theorems rely on the *Kullback-Leibler* divergence $\mathrm{KL}(\hat{\rho} \| \pi) \;=\; \mathbf{E}_{f \sim \hat{\rho}} \ln[\hat{\rho}(f)/\pi(f)]$ between a *prior* distribution $\pi$ over $\mathcal{F}$—specified before seeing the learning sample $X, Y$—and the posterior $\hat{\rho}$—typically obtained by feeding a learning process with $(X, Y)$.

Two appealing aspects of PAC-Bayesian theorems are that they provide data-driven generalization bounds that are computed on the training sample (*i.e.*, they do not rely on a testing sample), and that they are uniformly valid for all $\hat{\rho}$ over $\mathcal{F}$. This explains why many works study them as model selection criteria or as an inspiration for learning algorithm conception. Theorem 1, due to Catoni [8], has been used to derive or study learning algorithms [10, 17, 26, 27].

**Theorem 1** (Catoni [8]). *Given a distribution $\mathcal{D}$ over $\mathcal{X} \times \mathcal{Y}$, a hypothesis set $\mathcal{F}$, a loss function $\ell' : \mathcal{F} \times \mathcal{X} \times \mathcal{Y} \to [0, 1]$, a prior distribution $\pi$ over $\mathcal{F}$, a real number $\delta \in (0, 1]$, and a real number $\beta > 0$, with probability at least $1 - \delta$ over the choice of $(X, Y) \sim \mathcal{D}^n$, we have*

$$\forall \hat{\rho} \text{ on } \mathcal{F} : \quad \mathop{\mathbf{E}}_{f \sim \hat{\rho}} \mathcal{L}_{\mathcal{D}}^{\ell'}(f) \;\leq\; \frac{1}{1 - e^{-\beta}} \left[ 1 - e^{-\beta \, \mathbf{E}_{f \sim \hat{\rho}} \widehat{\mathcal{L}}_{X,Y}^{\ell'}(f) - \frac{1}{n} \left( \mathrm{KL}(\hat{\rho} \| \pi) + \ln \frac{1}{\delta} \right)} \right]. \tag{1}$$

Theorem 1 is limited to loss functions mapping to the range $[0, 1]$. Through a straightforward rescaling we can extend it to any bounded loss, *i.e.*, $\ell : \mathcal{F} \times \mathcal{X} \times \mathcal{Y} \to [a, b]$, where $[a, b] \subset \mathbb{R}$. This is done by using $\beta := b - a$ and with the *rescaled* loss function $\ell'(f, x, y) := (\ell(f, x, y) - a)/(b - a) \in [0, 1]$. After few arithmetic manipulations, we can rewrite Equation (1) as

$$\forall \hat{\rho} \text{ on } \mathcal{F} : \quad \mathop{\mathbf{E}}_{f \sim \hat{\rho}} \mathcal{L}_{\mathcal{D}}^{\ell}(f) \;\leq\; a + \frac{b - a}{1 - e^{a - b}} \left[ 1 - \exp \left( - \mathop{\mathbf{E}}_{f \sim \hat{\rho}} \widehat{\mathcal{L}}_{X,Y}^{\ell}(f) + a - \frac{1}{n} \left( \mathrm{KL}(\hat{\rho} \| \pi) + \ln \frac{1}{\delta} \right) \right) \right]. \tag{2}$$

From an algorithm design perspective, Equation (2) suggests optimizing a trade-off between the empirical expected loss and the Kullback-Leibler divergence. Indeed, for fixed $\pi, X, Y, n$, and $\delta$, minimizing Equation (2) is equivalent to find the distribution $\hat{\rho}$ that minimizes

$$n \mathop{\mathbf{E}}_{f \sim \hat{\rho}} \widehat{\mathcal{L}}_{X,Y}^{\ell}(f) + \mathrm{KL}(\hat{\rho} \| \pi) \,. \tag{3}$$

It is well known [1, 8, 10, 21] that the *optimal Gibbs posterior* $\hat{\rho}^*$ is given by

$$\hat{\rho}^*(f) \;=\; \tfrac{1}{Z_{X,Y}} \pi(f) \, e^{-n \, \widehat{\mathcal{L}}_{X,Y}^{\ell}(f)} \,, \tag{4}$$

where $Z_{X,Y}$ is a normalization term. Notice that the constant $\beta$ of Equation (1) is now absorbed in the loss function as the rescaling factor setting the trade-off between the expected empirical loss and $\mathrm{KL}(\hat{\rho} \| \pi)$.

## 3 Bridging Bayes and PAC-Bayes

In this section, we show that by choosing the negative log-likelihood loss function, minimizing the PAC-Bayes bound is equivalent to maximizing the Bayesian marginal likelihood. To obtain this result, we first consider the Bayesian approach that starts by defining a prior $p(\theta)$ over the set of possible model parameters $\Theta$. This induces a set of probabilistic estimators $f_\theta \in \mathcal{F}$, mapping $x$ to a probability distribution over $\mathcal{Y}$. Then, we can estimate the likelihood of observing $y$ given $x$ and $\theta$, *i.e.*, $p(y|x, \theta) \equiv f_\theta(y|x)$.[2] Using Bayes' rule, we obtain the posterior $p(\theta|X, Y)$:

$$p(\theta|X, Y) \;=\; \frac{p(\theta) \, p(Y|X, \theta)}{p(Y|X)} \;\propto\; p(\theta) \, p(Y|X, \theta) \,, \tag{5}$$

where $p(Y|X, \theta) \;=\; \prod_{i=1}^{n} p(y_i|x_i, \theta)$ and $p(Y|X) \;=\; \int_{\Theta} p(\theta) \, p(Y|X, \theta) \, d\theta$.

To bridge the Bayesian approach with the PAC-Bayesian framework, we consider the *negative log-likelihood* loss function [3], denoted $\ell_{\mathrm{nll}}$ and defined by

$$\ell_{\mathrm{nll}}(f_\theta, x, y) \;\equiv\; -\ln p(y|x, \theta)\,. \tag{6}$$

Then, we can relate the *empirical loss* $\widehat{\mathcal{L}}_{X,Y}^{\ell}$ of a predictor to its likelihood:

$$\widehat{\mathcal{L}}_{X,Y}^{\ell_{\mathrm{nll}}}(\theta) \;=\; \frac{1}{n}\sum_{i=1}^{n} \ell_{\mathrm{nll}}(\theta, x_i, y_i) \;=\; -\frac{1}{n}\sum_{i=1}^{n} \ln p(y_i|x_i, \theta) \;=\; -\frac{1}{n}\ln p(Y|X, \theta)\,,$$

or, the other way around,

$$p(Y|X, \theta) \;=\; e^{-n\,\widehat{\mathcal{L}}_{X,Y}^{\ell_{\mathrm{nll}}}(\theta)}\,. \tag{7}$$

Unfortunately, existing PAC-Bayesian theorems work with bounded loss functions or in very specific contexts [*e.g.*, 9, 36], and $\ell_{\mathrm{nll}}$ spans the whole real axis in its general form. In Section 4, we explore PAC-Bayes bounds for unbounded losses. Meanwhile, we consider priors with bounded likelihood. This can be done by assigning a prior of zero to any $\theta$ yielding $\ln\frac{1}{p(y|x,\theta)} \notin [a,b]$.

Now, using Equation (7) in the optimal posterior (Equation 4) simplifies to

$$\hat{\rho}^*(\theta) = \frac{\pi(\theta)\, e^{-n\,\widehat{\mathcal{L}}_{X,Y}^{\ell_{\mathrm{nll}}}(\theta)}}{Z_{X,Y}} \;=\; \frac{p(\theta)\, p(Y|X,\theta)}{p(Y|X)} = p(\theta|X, Y)\,, \tag{8}$$

where the normalization constant $Z_{X,Y}$ corresponds to the Bayesian *marginal likelihood*:

$$Z_{X,Y} \;\equiv\; p(Y|X) \;=\; \int_\Theta \pi(\theta)\, e^{-n\,\widehat{\mathcal{L}}_{X,Y}^{\ell_{\mathrm{nll}}}(\theta)} d\theta\,. \tag{9}$$

This shows that the optimal PAC-Bayes posterior given by the generalization bound of Theorem 1 coincides with the Bayesian posterior, when one chooses $\ell_{\mathrm{nll}}$ as loss function and $\beta := b-a$ (as in Equation 2). Moreover, using the posterior of Equation (8) inside Equation (3), we obtain

$$n \,\mathop{\mathbf{E}}_{\theta\sim\hat{\rho}^*} \widehat{\mathcal{L}}_{X,Y}^{\ell_{\mathrm{nll}}}(\theta) + \mathrm{KL}(\hat{\rho}^*\|\pi) \tag{10}$$

$$= \; n\int_\Theta \frac{\pi(\theta)\, e^{-n\,\widehat{\mathcal{L}}_{X,Y}^{\ell_{\mathrm{nll}}}(\theta)}}{Z_{X,Y}} \widehat{\mathcal{L}}_{X,Y}^{\ell_{\mathrm{nll}}}(\theta)\, d\theta + \int_\Theta \frac{\pi(\theta)\, e^{-n\,\widehat{\mathcal{L}}_{X,Y}^{\ell_{\mathrm{nll}}}(\theta)}}{Z_{X,Y}} \ln\left[\frac{\pi(\theta)\, e^{-n\,\widehat{\mathcal{L}}_{X,Y}^{\ell_{\mathrm{nll}}}(\theta)}}{\pi(\theta)\, Z_{X,Y}}\right] d\theta$$

$$= \; \int_\Theta \frac{\pi(\theta)\, e^{-n\,\widehat{\mathcal{L}}_{X,Y}^{\ell_{\mathrm{nll}}}(\theta)}}{Z_{X,Y}}\left[\ln\frac{1}{Z_{X,Y}}\right] d\theta \;=\; \frac{Z_{X,Y}}{Z_{X,Y}}\ln\frac{1}{Z_{X,Y}} \;=\; -\ln Z_{X,Y}\,.$$

In other words, minimizing the PAC-Bayes bound is equivalent to maximizing the marginal likelihood. Thus, from the PAC-Bayesian standpoint, the latter encodes a trade-off between the averaged negative log-likelihood loss function and the prior-posterior Kullback-Leibler divergence. Note that Equation (10) has been mentioned by Grünwald [14], based on an earlier observation of Zhang [36]. However, the PAC-Bayesian theorems proposed by the latter do not bound the generalization loss directly, as the "classical" PAC-Bayesian results [8, 24, 29] that we extend to regression in forthcoming Section 4 (see the corresponding remarks in Appendix A.1).

We conclude this section by proposing a compact form of Theorem 1 by expressing it in terms of the marginal likelihood, as a direct consequence of Equation (10).

**Corollary 2.** *Given a data distribution $\mathcal{D}$, a parameter set $\Theta$, a prior distribution $\pi$ over $\Theta$, a $\delta \in (0,1]$, if $\ell_{\mathrm{nll}}$ lies in $[a,b]$, we have, with probability at least $1-\delta$ over the choice of $(X,Y)\sim\mathcal{D}^n$,*

$$\mathop{\mathbf{E}}_{\theta\sim\hat{\rho}^*} \mathcal{L}_{\mathcal{D}}^{\ell_{\mathrm{nll}}}(\theta) \;\leq\; a + \tfrac{b-a}{1-e^{a-b}}\left[1 - e^a\,\sqrt[n]{Z_{X,Y}\,\delta}\,\right],$$

*where $\hat{\rho}^*$ is the Gibbs optimal posterior (Eq. 8) and $Z_{X,Y}$ is the marginal likelihood (Eq. 9).*

In Section 5, we exploit the link between PAC-Bayesian bounds and Bayesian marginal likelihood to expose similarities between both frameworks in the context of model selection. Beforehand, next Section 4 extends the PAC-Bayesian generalization guarantees to unbounded loss functions. This is mandatory to make our study fully valid, as the negative log-likelihood loss function is in general unbounded (as well as other common regression losses).

# 4 PAC-Bayesian Bounds for Regression

This section aims to extend the PAC-Bayesian results of Section 3 to real valued unbounded loss. These results are used in forthcoming sections to study $\ell_{\mathrm{nll}}$, but they are valid for broader classes of loss functions. Importantly, our new results are focused on regression problems, as opposed to the usual PAC-Bayesian classification framework.

The new bounds are obtained through a recent theorem of Alquier et al. [1], stated below (we provide a proof in Appendix A.2 for completeness).

**Theorem 3** (Alquier et al. [1]). *Given a distribution $\mathcal{D}$ over $\mathcal{X} \times \mathcal{Y}$, a hypothesis set $\mathcal{F}$, a loss function $\ell : \mathcal{F} \times \mathcal{X} \times \mathcal{Y} \to \mathbb{R}$, a prior distribution $\pi$ over $\mathcal{F}$, a $\delta \in (0,1]$, and a real number $\lambda > 0$, with probability at least $1-\delta$ over the choice of $(X,Y) \sim \mathcal{D}^n$, we have*

$$\forall \hat{\rho} \text{ on } \mathcal{F}: \quad \mathop{\mathbf{E}}_{f \sim \hat{\rho}} \mathcal{L}_{\mathcal{D}}^{\ell}(f) \leq \mathop{\mathbf{E}}_{f \sim \hat{\rho}} \widehat{\mathcal{L}}_{X,Y}^{\ell}(f) + \frac{1}{\lambda}\left[\mathrm{KL}(\hat{\rho}\|\pi) + \ln\frac{1}{\delta} + \Psi_{\ell,\pi,\mathcal{D}}(\lambda,n)\right], \quad (11)$$

$$\text{where} \quad \Psi_{\ell,\pi,\mathcal{D}}(\lambda,n) = \ln \mathop{\mathbf{E}}_{f \sim \pi} \mathop{\mathbf{E}}_{X',Y' \sim \mathcal{D}^n} \exp\left[\lambda\left(\mathcal{L}_{\mathcal{D}}^{\ell}(f) - \widehat{\mathcal{L}}_{X',Y'}^{\ell}(f)\right)\right]. \quad (12)$$

Alquier et al. used Theorem 3 to design a learning algorithm for $\{0,1\}$-valued classification losses. Indeed, a bounded loss function $\ell : \mathcal{F} \times \mathcal{X} \times \mathcal{Y} \to [a,b]$ can be used along with Theorem 3 by applying the Hoeffding's lemma to Equation (12), that gives $\Psi_{\ell,\pi,\mathcal{D}}(\lambda,n) \leq \lambda^2(b-a)^2/(2n)$. More specifically, with $\lambda := n$, we obtain the following bound

$$\forall \hat{\rho} \text{ on } \mathcal{F}: \quad \mathop{\mathbf{E}}_{f \sim \hat{\rho}} \mathcal{L}_{\mathcal{D}}^{\ell}(f) \leq \mathop{\mathbf{E}}_{f \sim \hat{\rho}} \widehat{\mathcal{L}}_{X,Y}^{\ell}(f) + \frac{1}{n}\left[\mathrm{KL}(\hat{\rho}\|\pi) + \ln\frac{1}{\delta}\right] + \frac{1}{2}(b-a)^2. \quad (13)$$

Note that the latter bound leads to the same trade-off as Theorem 1 (expressed by Equation 3). However, the choice $\lambda := n$ has the inconvenience that the bound value is at least $\frac{1}{2}(b-a)^2$, even at the limit $n \to \infty$. With $\lambda := \sqrt{n}$ the bound converges (a result similar to Equation (14) is also formulated by Pentina and Lampert [28]):

$$\forall \hat{\rho} \text{ on } \mathcal{F}: \quad \mathop{\mathbf{E}}_{f \sim \hat{\rho}} \mathcal{L}_{\mathcal{D}}^{\ell}(f) \leq \mathop{\mathbf{E}}_{f \sim \hat{\rho}} \widehat{\mathcal{L}}_{X,Y}^{\ell}(f) + \frac{1}{\sqrt{n}}\left[\mathrm{KL}(\hat{\rho}\|\pi) + \ln\frac{1}{\delta} + \frac{1}{2}(b-a)^2\right]. \quad (14)$$

**Sub-Gaussian losses.** In a regression context, it may be restrictive to consider strictly bounded loss functions. Therefore, we extend Theorem 3 to *sub-Gaussian* losses. We say that a loss function $\ell$ is sub-Gaussian with variance factor $s^2$ under a prior $\pi$ and a data-distribution $\mathcal{D}$ if it can be described by a sub-Gaussian random variable $V = \mathcal{L}_{\mathcal{D}}^{\ell}(f) - \ell(f,x,y)$, *i.e.*, its moment generating function is upper bounded by the one of a normal distribution of variance $s^2$ (see Boucheron et al. [7, Section 2.3]):

$$\psi_V(\lambda) = \ln \mathbf{E}\, e^{\lambda V} = \ln \mathop{\mathbf{E}}_{f \sim \pi} \mathop{\mathbf{E}}_{(x,y) \sim \mathcal{D}} \exp\left[\lambda\left(\mathcal{L}_{\mathcal{D}}^{\ell}(f) - \ell(f,x,y)\right)\right] \leq \frac{\lambda^2 s^2}{2}, \quad \forall \lambda \in \mathbb{R}. \quad (15)$$

The above sub-Gaussian assumption corresponds to the *Hoeffding assumption* of Alquier et al. [1], and allows to obtain the following result.

**Corollary 4.** *Given $\mathcal{D}$, $\mathcal{F}$, $\ell$, $\pi$ and $\delta$ defined in the statement of Theorem 3, if the loss is sub-Gaussian with variance factor $s^2$, we have, with probability at least $1-\delta$ over the choice of $(X,Y) \sim \mathcal{D}^n$,*

$$\forall \hat{\rho} \text{ on } \mathcal{F}: \quad \mathop{\mathbf{E}}_{f \sim \hat{\rho}} \mathcal{L}_{\mathcal{D}}^{\ell}(f) \leq \mathop{\mathbf{E}}_{f \sim \hat{\rho}} \widehat{\mathcal{L}}_{X,Y}^{\ell}(f) + \frac{1}{n}\left[\mathrm{KL}(\hat{\rho}\|\pi) + \ln\frac{1}{\delta}\right] + \frac{1}{2}s^2.$$

*Proof.* For $i = 1 \ldots n$, we denote $\ell_i$ a *i.i.d.* realization of the random variable $\mathcal{L}_{\mathcal{D}}^{\ell}(f) - \ell(f,x,y)$.

$\Psi_{\ell,\pi,\mathcal{D}}(\lambda,n) = \ln \mathbf{E} \exp\left[\frac{\lambda}{n}\sum_{i=1}^{n}\ell_i\right] = \ln \prod_{i=1}^{n} \mathbf{E}\exp\left[\frac{\lambda}{n}\ell_i\right] = \sum_{i=1}^{n}\psi_{\ell_i}(\frac{\lambda}{n}) \leq n\frac{\lambda^2 s^2}{2n^2} = \frac{\lambda^2 s^2}{2n}$,

where the inequality comes from the sub-Gaussian loss assumption (Equation 15). The result is then obtained from Theorem 3, with $\lambda := n$. $\qquad\square$

**Sub-gamma losses.** We say that an unbounded loss function $\ell$ is sub-gamma with a variance factor $s^2$ and scale parameter $c$, under a prior $\pi$ and a data-distribution $\mathcal{D}$, if it can be described by a sub-gamma random variable $V$ (see Boucheron et al. [7, Section 2.4]), that is

$$\psi_V(\lambda) \leq \frac{s^2}{c^2}(-\ln(1-\lambda c) - \lambda c) \leq \frac{\lambda^2 s^2}{2(1-c\lambda)}, \quad \forall \lambda \in (0, \tfrac{1}{c}). \quad (16)$$

Under this sub-gamma assumption, we obtain the following new result, which is necessary to study linear regression in the next sections.

**Corollary 5.** *Given $\mathcal{D}$, $\mathcal{F}$, $\ell$, $\pi$ and $\delta$ defined in the statement of Theorem 3, if the loss is sub-gamma with variance factor $s^2$ and scale $c < 1$, we have, with probability at least $1-\delta$ over $(X, Y) \sim \mathcal{D}^n$,*

$$\forall \hat{\rho} \text{ on } \mathcal{F}: \quad \mathop{\mathbf{E}}_{f \sim \hat{\rho}} \mathcal{L}_{\mathcal{D}}^{\ell}(f) \ \leq \ \mathop{\mathbf{E}}_{f \sim \hat{\rho}} \widehat{\mathcal{L}}_{X,Y}^{\ell}(f) + \tfrac{1}{n}\left[\mathrm{KL}(\hat{\rho}\|\pi) + \ln \tfrac{1}{\delta}\right] + \tfrac{1}{2(1-c)}\, s^2 \,. \tag{17}$$

*As a special case, with $\ell := \ell_{\mathrm{nll}}$ and $\hat{\rho} := \hat{\rho}^*$ (Equation 8), we have*

$$\mathop{\mathbf{E}}_{\theta \sim \hat{\rho}^*} \mathcal{L}_{\mathcal{D}}^{\ell_{\mathrm{nll}}}(\theta) \ \leq \ \tfrac{s^2}{2(1-c)} - \tfrac{1}{n} \ln\left(Z_{X,Y}\, \delta\right). \tag{18}$$

*Proof.* Following the same path as in the proof of Corollary 4 (with $\lambda := n$), we have

$$\Psi_{\ell,\pi,\mathcal{D}}(n,n) = \ln \mathbf{E} \exp\left[\textstyle\sum_{i=1}^n \ell_i\right] = \ln \prod_{i=1}^n \mathbf{E} \exp\left[\ell_i\right] = \textstyle\sum_{i=1}^n \psi_{\ell_i}(1) \ \leq \ n\tfrac{s^2}{2(1-c)} \ = \ \tfrac{n\,s^2}{2(1-c)}\,,$$

where the inequality comes from the sub-gamma loss assumption, with $1 \in (0, \tfrac{1}{c})$. $\qquad\square$

**Squared loss.** The parameters $s$ and $c$ of Corollary 5 rely on the chosen loss function and prior, and the assumptions concerning the data distribution. As an example, consider a regression problem where $\mathcal{X} \times \mathcal{Y} \subset \mathbb{R}^d \times \mathbb{R}$, a family of linear predictors $f_{\mathbf{w}}(\mathbf{x}) = \mathbf{w} \cdot \mathbf{x}$, with $\mathbf{w} \in \mathbb{R}^d$, and a Gaussian prior $\mathcal{N}(\mathbf{0}, \sigma_\pi^2\, \mathbf{I})$. Let us assume that the input examples are generated by $\mathbf{x} \sim \mathcal{N}(\mathbf{0}, \sigma_{\mathbf{x}}^2\, \mathbf{I})$ with label $y = \mathbf{w}^* \cdot \mathbf{x} + \epsilon$, where $\mathbf{w}^* \in \mathbb{R}^d$ and $\epsilon \sim \mathcal{N}(0, \sigma_\epsilon^2)$ is a Gaussian noise. Under the squared loss function

$$\ell_{\mathrm{sqr}}(\mathbf{w}, \mathbf{x}, y) \ = \ (\mathbf{w} \cdot \mathbf{x} - y)^2 \,, \tag{19}$$

we show in Appendix A.4 that Corollary 5 is valid with $s^2 \geq 2\left[\sigma_{\mathbf{x}}^2(\sigma_\pi^2 d + \|\mathbf{w}^*\|^2) + \sigma_\epsilon^2(1 - c)\right]$ and $c \geq 2\sigma_{\mathbf{x}}^2\sigma_\pi^2$. As expected, the bound degrades when the noise increases

**Regression versus classification.** The classical PAC-Bayesian theorems are stated in a classification context and bound the generalization error/loss of the stochastic *Gibbs predictor* $G_{\hat{\rho}}$. In order to predict the label of an example $x \in \mathcal{X}$, the Gibbs predictor first draws a hypothesis $h \in \mathcal{F}$ according to $\hat{\rho}$, and then returns $h(x)$. Maurer [23] shows that we can generalize PAC-Bayesian bounds on the generalization risk of the Gibbs classifier to any loss function with output between zero and one. Provided that $y \in \{-1, 1\}$ and $h(x) \in [-1, 1]$, a common choice is to use the linear loss function $\ell_{01}'(h, x, y) = \frac{1}{2} - \frac{1}{2}y\,h(x)$. The Gibbs generalization loss is then given by $R_{\mathcal{D}}(G_{\hat{\rho}}) = \mathbf{E}_{(x,y) \sim \mathcal{D}} \mathbf{E}_{h \sim \hat{\rho}} \ell_{01}'(h, x, y)$. Many PAC-Bayesian works use $R_{\mathcal{D}}(G_{\hat{\rho}})$ as a surrogate loss to study the zero-one classification loss of the majority vote classifier $R_{\mathcal{D}}(B_{\hat{\rho}})$:

$$R_{\mathcal{D}}(B_{\hat{\rho}}) \ = \ \Pr_{(x,y) \sim \mathcal{D}}\left(y \mathop{\mathbf{E}}_{h \sim \hat{\rho}} h(x) < 0\right) \ = \ \mathop{\mathbf{E}}_{(x,y) \sim \mathcal{D}} I\left[y \mathop{\mathbf{E}}_{h \sim \hat{\rho}} h(x) < 0\right], \tag{20}$$

where $I[\cdot]$ being the indicator function. Given a distribution $\hat{\rho}$, an upper bound on the Gibbs risk is converted to an upper bound on the majority vote risk by $R_{\mathcal{D}}(B_{\hat{\rho}}) \leq 2R_{\mathcal{D}}(G_{\hat{\rho}})$ [20]. In some situations, this *factor of two* may be reached, *i.e.*, $R_{\mathcal{D}}(B_{\hat{\rho}}) \simeq 2R_{\mathcal{D}}(G_{\hat{\rho}})$. In other situations, we may have $R_{\mathcal{D}}(B_{\hat{\rho}}) = 0$ even if $R_{\mathcal{D}}(G_{\hat{\rho}}) = \frac{1}{2}-\epsilon$ (see Germain et al. [11] for an extensive study). Indeed, these bounds obtained via the Gibbs risk are exposed to be loose and/or unrepresentative of the majority vote generalization error.[3]

In the current work, we study regression losses instead of classification ones. That is, the provided results express upper bounds on $\mathbf{E}_{f \sim \hat{\rho}} \mathcal{L}_{\mathcal{D}}^{\ell}(f)$ for any (bounded, sub-Gaussian, or sub-gamma) losses. Of course, one may want to bound the regression loss of the averaged regressor $F_{\hat{\rho}}(x) = \mathbf{E}_{f \sim \hat{\rho}} f(x)$. In this case, if the loss function $\ell$ is convex (as the squared loss), Jensen's inequality gives $\mathcal{L}_{\mathcal{D}}^{\ell}(F_{\hat{\rho}}) \leq \mathbf{E}_{f \sim \hat{\rho}} \mathcal{L}_{\mathcal{D}}^{\ell}(f)$. Note that a strict inequality replaces the factor two mentioned above for the classification case, due to the non-convex indicator function of Equation (20).

Now that we have generalization bounds for real-valued loss functions, we can continue our study linking PAC-Bayesian results to Bayesian inference. In the next section, we focus on model selection.

## 5 Analysis of Model Selection

We consider $L$ distinct models $\{\mathcal{M}_i\}_{i=1}^L$, each one defined by a set of parameters $\Theta_i$. The PAC-Bayesian theorems naturally suggest selecting the model that is best adapted for the given task by evaluating the bound for each model $\{\mathcal{M}_i\}_{i=1}^L$ and selecting the one with the lowest bound [2, 25, 36]. This is closely linked with the Bayesian model selection procedure, as we showed in Section 3 that minimizing the PAC-Bayes bound amounts to maximizing the marginal likelihood. Indeed, given a collection of $L$ optimal Gibbs posteriors—one for each model—given by Equation (8),

$$p(\theta|X,Y,\mathcal{M}_i) \equiv \hat{\rho}_i^*(\theta) = \tfrac{1}{Z_{X,Y,i}}\pi_i(\theta)\,e^{-n\,\widehat{\mathcal{L}}_{X,Y}^{\ell_{\mathrm{nll}}}(\theta)}, \quad \text{for } \theta \in \Theta_i\,, \qquad (21)$$

the Bayesian Occam's razor criteria [18, 22] chooses the one with the higher *model evidence*

$$p(Y|X,\mathcal{M}_i) \equiv Z_{X,Y,i} = \int_{\Theta_i} \pi_i(\theta)\,e^{-n\,\widehat{\mathcal{L}}_{X,Y}^{\ell}(\theta)}\,d\theta\,. \qquad (22)$$

Corollary 6 below formally links the PAC-Bayesian and the Bayesian model selection. To obtain this result, we simply use the bound of Corollary 5 $L$ times, together with $\ell_{\mathrm{nll}}$ and Equation (10). From the union bound (*a.k.a.* Bonferroni inequality), it is mandatory to compute each bound with a confidence parameter of $\delta/L$, to ensure that the final conclusion is valid with probability at least $1-\delta$.

**Corollary 6.** *Given a data distribution $\mathcal{D}$, a family of model parameters $\{\Theta_i\}_{i=1}^L$ and associated priors $\{\pi_i\}_{i=1}^L$—where $\pi_i$ is defined over $\Theta_i$—, a $\delta \in (0,1]$, if the loss is sub-gamma with parameters $s^2$ and $c < 1$, then, with probability at least $1 - \delta$ over $(X,Y) \sim \mathcal{D}^n$,*

$$\forall i \in \{1,\ldots,L\} : \qquad \underset{\theta \sim \hat{\rho}_i^*}{\mathbf{E}} \mathcal{L}_{\mathcal{D}}^{\ell_{\mathrm{nll}}}(\theta) \leq \tfrac{1}{2(1-c)}\,s^2 - \tfrac{1}{n}\ln\left(Z_{X,Y,i}\,\tfrac{\delta}{L}\right).$$

*where $\hat{\rho}_i^*$ is the Gibbs optimal posterior (Eq. 21) and $Z_{X,Y,i}$ is the marginal likelihood (Eq. 22).*

Hence, under the uniform prior over the $L$ models, choosing the one with the best model evidence is equivalent to choosing the one with the lowest PAC-Bayesian bound.

**Hierarchical Bayes.** To perform proper inference on hyperparameters, we have to rely on the *Hierarchical Bayes* approach. This is done by considering an *hyperprior* $p(\eta)$ over the set of hyperparameters H. Then, the prior $p(\theta|\eta)$ can be conditioned on a choice of hyperparameter $\eta$. The Bayes rule of Equation (5) becomes $p(\theta,\eta|X,Y) = \frac{p(\eta)\,p(\theta|\eta)\,p(Y|X,\theta)}{p(Y|X)}$.

Under the negative log-likelihood loss function, we can rewrite the results of Corollary 5 as a generalization bound on $\mathbf{E}_{\eta \sim \hat{\rho}_0} \mathbf{E}_{\theta \sim \hat{\rho}_\eta^*} \mathcal{L}_{\mathcal{D}}^{\ell_{\mathrm{nll}}}(\theta)$, where $\hat{\rho}_0(\eta) \propto \pi_0(\eta)\,Z_{X,Y,\eta}$ is the hyperposterior on H and $\pi_0$ the hyperprior. Indeed, Equation (18) becomes

$$\underset{\theta \sim \hat{\rho}^*}{\mathbf{E}} \mathcal{L}_{\mathcal{D}}^{\ell_{\mathrm{nll}}}(\theta) = \underset{\eta \sim \hat{\rho}_0^*}{\mathbf{E}}\,\underset{\theta \sim \hat{\rho}_\eta^*}{\mathbf{E}} \mathcal{L}_{\mathcal{D}}^{\ell_{\mathrm{nll}}}(\theta) \leq \tfrac{1}{2(1-c)}\,s^2 - \tfrac{1}{n}\ln\left(\underset{\eta \sim \pi_0}{\mathbf{E}} Z_{X,Y,\eta}\,\delta\right). \qquad (23)$$

To relate to the bound obtained in Corollary 6, we consider the case of a discrete hyperparameter set $H = \{\eta_i\}_{i=1}^L$, with a uniform prior $\pi_0(\eta_i) = \tfrac{1}{L}$ (from now on, we regard each hyperparameter $\eta_i$ as the specification of a model $\Theta_i$). Then, Equation (23) becomes

$$\underset{\theta \sim \hat{\rho}^*}{\mathbf{E}} \mathcal{L}_{\mathcal{D}}^{\ell_{\mathrm{nll}}}(\theta) = \underset{\eta \sim \hat{\rho}_0^*}{\mathbf{E}}\,\underset{\theta \sim \hat{\rho}_\eta^*}{\mathbf{E}} \mathcal{L}_{\mathcal{D}}^{\ell_{\mathrm{nll}}}(\theta) \leq \tfrac{1}{2(1-c)}\,s^2 - \tfrac{1}{n}\ln\left(\sum_{i=1}^L Z_{X,Y,\eta_i}\,\tfrac{\delta}{L}\right).$$

This bound is now a function of $\sum_{i=1}^L Z_{X,Y,\eta_i}$ instead of $\max_i Z_{X,Y,\eta_i}$ as in the bound given by the "best" model in Corollary 6. This yields a tighter bound, corroborating the Bayesian wisdom that model averaging performs best. Conversely, when selecting a single hyperparameter $\eta^* \in H$, the hierarchical representation is equivalent to choosing a deterministic hyperposterior, satisfying $\hat{\rho}_0(\eta^*) = 1$ and 0 for every other values. We then have

$$\mathrm{KL}(\hat{\rho}||\pi) = \mathrm{KL}(\hat{\rho}_0||\pi_0) + \underset{\eta \sim \hat{\rho}_0}{\mathbf{E}} \mathrm{KL}(\hat{\rho}_\eta||\pi_\eta) = \ln(L) + \mathrm{KL}(\hat{\rho}_{\eta^*}||\pi_{\eta^*})\,.$$

With the optimal posterior for the selected $\eta^*$, we have

$$n\underset{\theta \sim \hat{\rho}}{\mathbf{E}} \widehat{\mathcal{L}}_{X,Y}^{\ell_{\mathrm{nll}}}(\theta) + \mathrm{KL}(\hat{\rho}||\pi) = n\underset{\theta \sim \hat{\rho}_\eta^*}{\mathbf{E}} \widehat{\mathcal{L}}_{X,Y}^{\ell_{\mathrm{nll}}}(\theta) + \mathrm{KL}(\hat{\rho}_{\eta^*}^*||\pi_{\eta^*}) + \ln(L)$$
$$= -\ln(Z_{X,Y,\eta^*}) + \ln(L) = -\ln\left(\tfrac{Z_{X,Y,\eta^*}}{L}\right).$$

Inserting this result into Equation (17), we fall back on the bound obtained in Corollary 6. Hence, by comparing the values of the bounds, one can get an estimate on the consequence of performing model selection instead of model averaging.

# 6 Linear Regression

In this section, we perform *Bayesian linear regression* using the parameterization of Bishop [5]. The output space is $\mathcal{Y} := \mathbb{R}$ and, for an arbitrary input space $\mathcal{X}$, we use a mapping function $\boldsymbol{\phi} : \mathcal{X} \to \mathbb{R}^d$.

**The model.** Given $(x, y) \in \mathcal{X} \times \mathcal{Y}$ and model parameters $\theta := \langle \mathbf{w}, \sigma \rangle \in \mathbb{R}^d \times \mathbb{R}^+$, we consider the likelihood $p(y|x, \langle \mathbf{w}, \sigma \rangle) = \mathcal{N}(y|\mathbf{w} \cdot \boldsymbol{\phi}(\mathbf{x}), \sigma^2)$. Thus, the negative log-likelihood loss is

$$\ell_{\text{nll}}(\langle \mathbf{w}, \sigma \rangle, x, y) = -\ln p(y|x, \langle \mathbf{w}, \sigma \rangle) = \tfrac{1}{2}\ln(2\pi\sigma^2) + \tfrac{1}{2\sigma^2}(y - \mathbf{w} \cdot \boldsymbol{\phi}(x))^2 \,. \tag{24}$$

For a fixed $\sigma^2$, minimizing Equation (24) is equivalent to minimizing the squared loss function of Equation (19). We also consider an isotropic Gaussian prior of mean $\mathbf{0}$ and variance $\sigma_\pi^2$: $p(\mathbf{w}|\sigma_\pi) = \mathcal{N}(\mathbf{w}|\mathbf{0}, \sigma_\pi^2 \mathbf{I})$. For the sake of simplicity, we consider fixed parameters $\sigma^2$ and $\sigma_\pi^2$. The Gibbs optimal posterior (see Equation 8) is then given by

$$\hat{\rho}^*(\mathbf{w}) \equiv p(\mathbf{w}|X, Y, \sigma, \sigma_\pi) = \tfrac{p(\mathbf{w}|\sigma_\pi)\,p(Y|X,\mathbf{w},\sigma)}{p(Y|X,\sigma,\sigma_\pi)} = \mathcal{N}(\mathbf{w}\,|\,\widehat{\mathbf{w}}, A^{-1})\,, \tag{25}$$

where $A := \frac{1}{\sigma^2}\boldsymbol{\Phi}^T\boldsymbol{\Phi} + \frac{1}{\sigma_\pi^2}\mathbf{I}$ ; $\widehat{\mathbf{w}} := \frac{1}{\sigma^2}A^{-1}\boldsymbol{\Phi}^T\mathbf{y}$ ; $\boldsymbol{\Phi}$ is a $n \times d$ matrix such that the $i^{th}$ line is $\boldsymbol{\phi}(x_i)$ ; $\mathbf{y} := [y_1, \ldots y_n]$ is the labels-vector ; and the negative log marginal likelihood is

$$-\ln p(Y|X, \sigma, \sigma_\pi) = \tfrac{1}{2\sigma^2}\|\mathbf{y} - \boldsymbol{\Phi}\widehat{\mathbf{w}}\|^2 + \tfrac{n}{2}\ln(2\pi\sigma^2) + \tfrac{1}{2\sigma_\pi^2}\|\widehat{\mathbf{w}}\|^2 + \tfrac{1}{2}\log|A| + d\ln\sigma_\pi$$

$$= \underbrace{n\,\widehat{\mathcal{L}}_{X,Y}^{\ell_{\text{nll}}}(\widehat{\mathbf{w}}) + \tfrac{1}{2\sigma^2}\operatorname{tr}(\boldsymbol{\Phi}^T\boldsymbol{\Phi}A^{-1})}_{n\,\mathbf{E}_{\mathbf{w}\sim\hat{\rho}^*}\,\widehat{\mathcal{L}}_{X,Y}^{\ell_{\text{nll}}}(\mathbf{w})} + \underbrace{\tfrac{1}{2\sigma_\pi^2}\operatorname{tr}(A^{-1}) - \tfrac{d}{2} + \tfrac{1}{2\sigma_\pi^2}\|\widehat{\mathbf{w}}\|^2 + \tfrac{1}{2}\log|A| + d\ln\sigma_\pi}_{\text{KL}\left(\mathcal{N}(\widehat{\mathbf{w}},A^{-1})\,\|\,\mathcal{N}(\mathbf{0},\sigma_\pi^2\mathbf{I})\right)} \,.$$

To obtain the second equality, we substitute $\frac{1}{2\sigma^2}\|\mathbf{y} - \boldsymbol{\Phi}\widehat{\mathbf{w}}\|^2 + \frac{n}{2}\ln(2\pi\sigma^2) = n\,\widehat{\mathcal{L}}_{X,Y}^{\ell_{\text{nll}}}(\widehat{\mathbf{w}})$ and insert

$$\tfrac{1}{2\sigma^2}\operatorname{tr}(\boldsymbol{\Phi}^T\boldsymbol{\Phi}A^{-1}) + \tfrac{1}{2\sigma_\pi^2}\operatorname{tr}(A^{-1}) = \tfrac{1}{2}\operatorname{tr}(\tfrac{1}{\sigma^2}\boldsymbol{\Phi}^T\boldsymbol{\Phi}A^{-1} + \tfrac{1}{\sigma_\pi^2}A^{-1}) = \tfrac{1}{2}\operatorname{tr}(A^{-1}A) = \tfrac{d}{2}\,.$$

This exhibits how the Bayesian regression optimization problem is related to the minimization of a PAC-Bayesian bound, expressed by a trade-off between $\mathbf{E}_{\mathbf{w}\sim\hat{\rho}^*}\,\widehat{\mathcal{L}}_{X,Y}^{\ell_{\text{nll}}}(\mathbf{w})$ and $\text{KL}\left(\mathcal{N}(\widehat{\mathbf{w}}, A^{-1})\,\|\,\mathcal{N}(\mathbf{0}, \sigma_\pi^2\mathbf{I})\right)$. See Appendix A.5 for detailed calculations.

**Model selection experiment.** To produce Figures 1a and 1b, we reimplemented the toy experiment of Bishop [5, Section 3.5.1]. That is, we generated a learning sample of 15 data points according to $y = \sin(x) + \epsilon$, where $x$ is uniformly sampled in the interval $[0, 2\pi]$ and $\epsilon \sim \mathcal{N}(0, \frac{1}{4})$ is a Gaussian noise. We then learn seven different polynomial models applying Equation (25). More precisely, for a polynomial model of degree $d$, we map input $x \in \mathbb{R}$ to a vector $\boldsymbol{\phi}(x) = [1, x^1, x^2, \ldots, x^d] \in \mathbb{R}^{d+1}$, and we fix parameters $\sigma_\pi^2 = \frac{1}{0.005}$ and $\sigma^2 = \frac{1}{2}$. Figure 1a illustrates the seven learned models. Figure 1b shows the negative log marginal likelihood computed for each polynomial model, and is designed to reproduce Bishop [5, Figure 3.14], where it is explained that the marginal likelihood correctly indicates that the polynomial model of degree $d = 3$ is "the simplest model which gives a good explanation for the observed data". We show that this claim is well quantified by the trade-off intrinsic to our PAC-Bayesian approach: the complexity KL term keeps increasing with the parameter $d \in \{1, 2, \ldots, 7\}$, while the empirical risk drastically decreases from $d = 2$ to $d = 3$, and only slightly afterward. Moreover, we show that the generalization risk (computed on a test sample of size 1000) tends to increase with complex models (for $d \geq 4$).

**Empirical comparison of bound values.** Figure 1c compares the values of the PAC-Bayesian bounds presented in this paper on a synthetic dataset, where each input $\mathbf{x} \in \mathbb{R}^{20}$ is generated by a Gaussian $\mathbf{x} \sim \mathcal{N}(\mathbf{0}, \mathbf{I})$. The associated output $y \in \mathbb{R}$ is given by $y = \mathbf{w}^* \cdot \mathbf{x} + \epsilon$, with $\|\mathbf{w}^*\| = \frac{1}{2}$, $\epsilon \sim \mathcal{N}(0, \sigma_\epsilon^2)$, and $\sigma_\epsilon^2 = \frac{1}{9}$. We perform Bayesian linear regression in the input space, *i.e.*, $\boldsymbol{\phi}(\mathbf{x}) = \mathbf{x}$, fixing $\sigma_\pi^2 = \frac{1}{100}$ and $\sigma^2 = 2$. That is, we compute the posterior of Equation (25) for training samples of sizes from 10 to $10^6$. For each learned model, we compute the empirical negative log-likelihood loss of Equation (24), and the three PAC-Bayes bounds, with confidence parameter of $\delta = \frac{1}{20}$. Note that this loss function is an affine transformation of the squared loss studied in Section 4 (Equation 19), *i.e.*, $\ell_{\text{nll}}(\langle \mathbf{w}, \sigma \rangle, \mathbf{x}, y) = \frac{1}{2}\ln(2\pi\sigma^2) + \frac{1}{2\sigma^2}\ell_{\text{sqr}}(\mathbf{w}, \mathbf{x}, y)$. It turns out that $\ell_{\text{nll}}$ is sub-gamma with parameters $s^2 \geq \frac{1}{\sigma^2}\left[\sigma_{\mathbf{x}}^2(\sigma_\pi^2 d + \|\mathbf{w}^*\|^2) + \sigma_\epsilon^2(1-c)\right]$ and $c \geq \frac{1}{\sigma^2}(\sigma_{\mathbf{x}}^2\sigma_\pi^2)$, as shown in Appendix A.6. The bounds of Corollary 5 are computed using the above mentioned values of $\|\mathbf{w}^*\|, d, \sigma, \sigma_{\mathbf{x}}, \sigma_\epsilon, \sigma_\pi$, leading

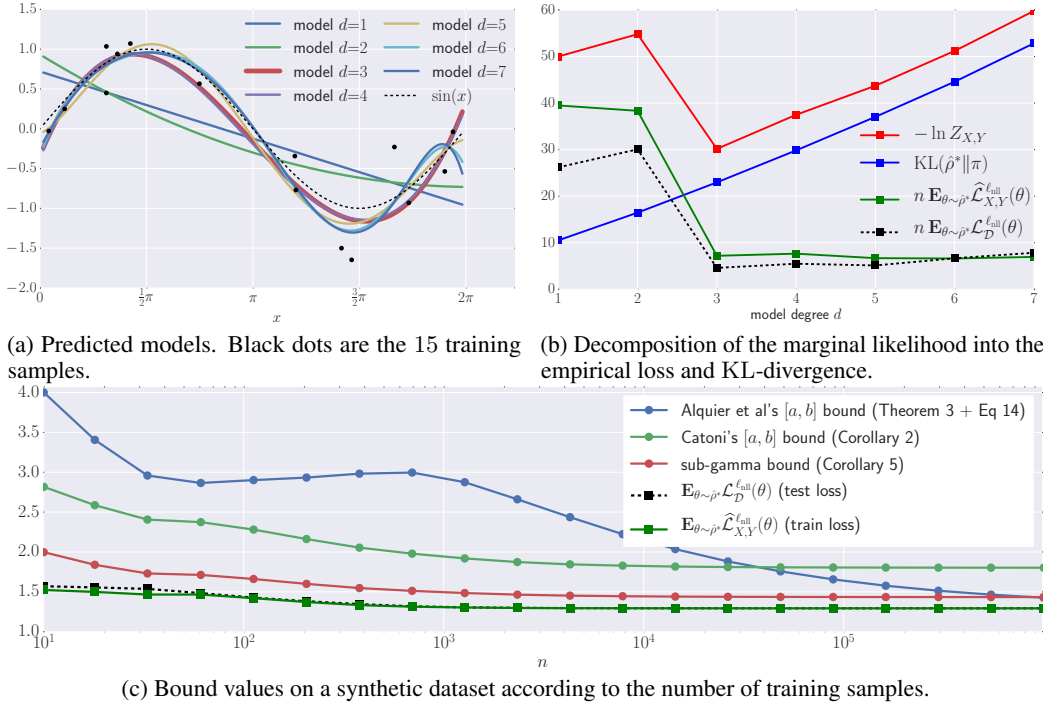

(a) Predicted models. Black dots are the 15 training samples.

(b) Decomposition of the marginal likelihood into the empirical loss and KL-divergence.

(c) Bound values on a synthetic dataset according to the number of training samples.

Figure 1: Model selection experiment (a-b); and comparison of bounds values (c).

to $s^2 \simeq 0.280$ and $c \simeq 0.005$. As the two other bounds of Figure 1c are not suited for unbounded loss, we compute their value using a cropped loss $[a, b] = [1, 4]$. Different parameter values could have been chosen, sometimes leading to another picture: a large value of $s$ degrades our sub-gamma bound, as a larger $[a, b]$ interval does for the other bounds.

In the studied setting, the bound of Corollary 5—that we have developed for (unbounded) sub-gamma losses—gives tighter guarantees than the two results for $[a, b]$-bounded losses (up to $n = 10^6$). However, our new bound always maintains a gap of $\frac{1}{2(1-c)}s^2$ between its value and the generalization loss. The result of Corollary 2 (adapted from Catoni [8]) for bounded losses suffers from a similar gap, while having higher values than our sub-gamma result. Finally, the result of Theorem 3 (Alquier et al. [1]), combined with $\lambda = 1/\sqrt{n}$ (Eq. 14), converges to the expected loss, but it provides good guarantees only for large training sample ($n \gtrsim 10^5$). Note that the latter bound is not directly minimized by our "optimal posterior", as opposed to the one with $\lambda = 1/n$ (Eq. 13), for which we observe values between 5.8 (for $n = 10^6$) and 6.4 (for $n = 10$)—not displayed on Figure 1c.

## 7 Conclusion

The first contribution of this paper is to bridge the concepts underlying the Bayesian and the PAC-Bayesian approaches; under proper parameterization, the minimization of the PAC-Bayesian bound maximizes the marginal likelihood. This study motivates the second contribution of this paper, which is to prove PAC-Bayesian generalization bounds for regression with unbounded sub-gamma loss functions, including the squared loss used in regression tasks.

In this work, we studied model selection techniques. On a broader perspective, we would like to suggest that both Bayesian and PAC-Bayesian frameworks may have more to learn from each other than what has been done lately (even if other works paved the way [e.g., 6, 14, 30]). Predictors learned from the Bayes rule can benefit from strong PAC-Bayesian frequentist guarantees (under the *i.i.d.* assumption). Also, the rich Bayesian toolbox may be incorporated in PAC-Bayesian driven algorithms and risk bounding techniques.

### Acknowledgments

We thank Gabriel Dubé and Maxime Tremblay for having proofread the paper and supplemental.

## Footnotes

[1]Some existing connections [3, 6, 14, 19, 29, 30, 36] are discussed in Appendix A.1.

[2]To stay aligned with the PAC-Bayesian setup, we only consider the discriminative case in this paper. One can extend to the generative setup by considering the likelihood of the form $p(y, x|\theta)$ instead.

[3]It is noteworthy that the best PAC-Bayesian empirical bound values are so far obtained by considering a majority vote of linear classifiers, where the prior and posterior are Gaussian [2, 10, 20], similarly to the Bayesian linear regression analyzed in Section 6.

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
