[Supplementary Material]

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

[4]The PAC-Bayesian results for Gaussian processes are summarized in Rasmussen and Williams [47, Section 7.4]

[5]The empirical model selection capabilities of the *Safe Bayesian* algorithm has been further studied in Grünwald and van Ommen [40].

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

# A  Supplementary Material

## A.1  Related Work

In this section, we discuss briefly other works containing (more or less indirect) links between Bayesian inference and PAC-Bayesian theory, and explain how they relate to the current paper.

**Seeger (2002, 2003) [29, 30].**   Soon after the initial work of McAllester [24, 25], Seeger shows how to apply the PAC-Bayesian theorems to bound the generalization error of Gaussian Processes in a classification context. By building upon the PAC-Bayesian theorem initially appearing in Langford and Seeger [44]—where the divergence between the training error and the generalization one is given by the Kullback-Leibler divergence between two Bernoulli distributions—it achieves very tight generalization bounds.[4] Also, the thesis of Seeger [30, Section 3.2] foresees this by noticing that "the log marginal likelihood incorporates a *similar trade-off* as the PAC-Bayesian theorem", but using another variant of the PAC-Bayes bound and in the context of classification.

**Banerjee (2006) [3].**   This paper shows similarities between the early PAC-Bayesian results (McAllester [25], Langford and Seeger [44]), and the *Bayesian log-loss bound* (Freund and Schapire [38], Kakade and Ng [42]). This is done by highlighting that the proof of all these results are strongly relying on the same *compression lemma* [3, Lemma 1], which is equivalent to our *change of measure* used in the proof of Theorem 3 (see forthcoming Equation 26). Note that the loss studied in the Bayesian part of Banerjee [3] is the negative log-likelihood of Equation (6). Also, as in Equation (10), the *Bayesian log-loss bound* contains the Kullback-Leibler divergence between the prior and the posterior. However, the latter result is not a generalization bound, but a bound on the training loss that is obtained by computing a surrogate training loss in the specific context of online learning. Moreover, the marginal likelihood and the model selection techniques are not addressed in Banerjee [3].

**Zhang (2006) [36].**   This paper presents a family of information theoretical bounds for *randomized estimators* that have a lot in common with PAC-Bayesian results (although the bounded quantity is not directly the generalization error). Minimizing these bounds leads to the same optimal Gibbs posterior of Equation (4). The author noted that using the negative log-likelihood (Equation 6) leads to the Bayesian posterior, but made no connection with the marginal likelihood.

**Grünwald (2012) [14].**   This paper proposes the *Safe Bayesian* algorithm, which selects a proper Bayesian *learning rate* — that is analogous to the parameter $\beta$ of our Equation (1), and the parameter $\lambda$ of our Equation (11) — in the context of *misspecified models*.[5] The standard Bayesian inference method is obtained with a fixed learning rate, corresponding to the case $\lambda := n$ (that is the case we focus on the current paper, see Corollaries 4 and 5). The analysis of Grünwald [14] relies both on the Minimum Description Length principle [41] and PAC-Bayesian theory. Building upon the work of Zhang [36] discussed above, they formulate the result that we presented as Equation (10), linking the marginal likelihood to the inherent PAC-Bayesian trade-off. However, they do not compute explicit bounds on the generalization loss, which required us to take into account the complexity term of Equation (12).

**Lacoste (2015) [19].**   In a binary classification context, it is shown that the parameter $\beta$ of Theorem 1 can be interpreted as a Bernoulli label noise model from a Bayesian likelihood standpoint. For more details, we refer the reader to Section 2.2 of this thesis.

**Bissiri et al. (2016) [6].**   This recent work studies Bayesian inference through the lens of loss functions. When the loss function is the negative log-likelihood (Equation 6), the approach of Bissiri et al. [6] coincides with the Bayesian update rule. As mentioned by the authors, there is some connection between their framework and the PAC-Bayesian one, but "the motivation and construction are very different."

**Other references.** See also Grünwald and Langford [39], Lacoste-Julien et al. [43], Meir and Zhang [45], Ng and Jordan [46], Rousseau [48] for other studies drawing links between frequentist statistics and Bayesian inference, but outside the PAC-Bayesian framework.

## A.2 Proof of Theorem 3

Recall that Theorem 3 originally comes from Alquier et al. [1, Theorem 4.1]. We present below a different proof that follows the key steps of the very general PAC-Bayesian theorem presented in Bégin et al. [37, Theorem 4].

*Proof of Theorem 3.* The *Donsker-Varadhan's change of measure* states that, for any measurable function $\phi : \mathcal{F} \to \mathbb{R}$, we have

$$\mathop{\mathbf{E}}_{f\sim\hat{\rho}} \phi(f) \ \leq \ \mathrm{KL}(\hat{\rho}\|\pi) + \ln\left(\mathop{\mathbf{E}}_{f\sim\pi} e^{\phi(f)}\right). \tag{26}$$

Thus, with $\phi(f) := \lambda\big(\mathcal{L}_{\mathcal{D}}^{\ell}(f) - \widehat{\mathcal{L}}_{X,Y}^{\ell}(f)\big)$, we obtain $\forall\,\hat{\rho}$ on $\mathcal{F}$ :

$$
\begin{aligned}
\lambda\Big( \mathop{\mathbf{E}}_{f\sim\hat{\rho}} \mathcal{L}_{\mathcal{D}}^{\ell}(f) - \mathop{\mathbf{E}}_{f\sim\hat{\rho}} \widehat{\mathcal{L}}_{X,Y}^{\ell}(f)\Big) \ &= \ \mathop{\mathbf{E}}_{f\sim\hat{\rho}} \lambda\left(\mathcal{L}_{\mathcal{D}}^{\ell}(f) - \widehat{\mathcal{L}}_{X,Y}^{\ell}(f)\right) \\
&\leq \ \mathrm{KL}(\hat{\rho}\|\pi) + \ln\left(\mathop{\mathbf{E}}_{f\sim\pi} e^{\lambda\left(\mathcal{L}_{\mathcal{D}}^{\ell}(f) - \widehat{\mathcal{L}}_{X,Y}^{\ell}(f)\right)}\right).
\end{aligned}
$$

Now, we apply Markov's inequality on the random variable $\zeta_{\pi}(X,Y) := \mathop{\mathbf{E}}_{f\sim\pi} e^{\lambda\left(\mathcal{L}_{\mathcal{D}}^{\ell}(f) - \widehat{\mathcal{L}}_{X,Y}^{\ell}(f)\right)}$:

$$\mathop{\mathrm{Pr}}_{X,Y\sim\mathcal{D}^n}\left(\zeta_{\pi}(X,Y) \leq \frac{1}{\delta}\mathop{\mathbf{E}}_{X',Y'\sim\mathcal{D}^n} \zeta_{\pi}(X',Y')\right) \geq 1-\delta\,.$$

This implies that with probability at least $1-\delta$ over the choice of $X, Y \sim \mathcal{D}^n$, we have $\forall\,\hat{\rho}$ on $\mathcal{F}$ :

$$\mathop{\mathbf{E}}_{f\sim\hat{\rho}} \mathcal{L}_{\mathcal{D}}^{\ell}(f) \ \leq \ \mathop{\mathbf{E}}_{f\sim\hat{\rho}} \widehat{\mathcal{L}}_{X,Y}^{\ell}(f) + \frac{1}{\lambda}\left[\mathrm{KL}(\hat{\rho}\|\pi) + \ln\frac{\mathop{\mathbf{E}}_{X',Y'\sim\mathcal{D}^n} \zeta_{\pi}(X',Y')}{\delta}\right].$$

$\square$

## A.3 Proof of Equations (13) and (14)

*Proof.* Given a loss function $\ell : \mathcal{F} \times \mathcal{X} \times \mathcal{Y}$, and a fixed predictor $f \in \mathcal{F}$, we consider the random experiment of sampling $(x, y) \in \mathcal{D}$. We denote $\ell_i$ a realization of the random variable $\mathcal{L}_{\mathcal{D}}^{\ell}(f) - \ell(f, x, y)$, for $i = 1 \ldots n$. Each $\ell_i$ is *i.i.d.*, zero mean, and bounded by $a - b$ and $b - a$, as $\ell(f, x, y) \in [a, b]$. Thus,

$$
\begin{aligned}
\mathop{\mathbf{E}}_{X',Y'\sim\mathcal{D}^n} \exp\left[\lambda\left(\mathcal{L}_{\mathcal{D}}^{\ell}(f) - \widehat{\mathcal{L}}_{X',Y'}^{\ell}(f)\right)\right] \ &= \ \mathbf{E}\exp\left[\frac{\lambda}{n}\sum_{i=1}^{n}\ell_i\right] \\
&= \ \prod_{i=1}^{n}\mathbf{E}\exp\left[\frac{\lambda}{n}\ell_i\right] \\
&\leq \ \prod_{i=1}^{n}\exp\left[\frac{\lambda^2(a - b - (b - a))^2}{8n^2}\right] \\
&= \ \prod_{i=1}^{n}\exp\left[\frac{\lambda^2(b - a)^2}{2n^2}\right] \\
&= \ \exp\left[\frac{\lambda^2(b - a)^2}{2n}\right],
\end{aligned}
$$

where the inequality comes from Hoeffding's lemma.

With $\lambda := n$, Equation (11) becomes Equation (13) :

$$\mathbf{E}_{f\sim\hat{\rho}} \mathcal{L}^{\ell}_{\mathcal{D}}(f) \;\leq\; \mathbf{E}_{f\sim\hat{\rho}} \widehat{\mathcal{L}}^{\ell}_{X,Y}(f) + \frac{1}{n}\left[\mathrm{KL}(\hat{\rho}\|\pi) + \ln\frac{1}{\delta} + \frac{n^2(b-a)^2}{2n}\right]$$

$$= \mathbf{E}_{f\sim\hat{\rho}} \widehat{\mathcal{L}}^{\ell}_{X,Y}(f) + \frac{1}{n}\left[\mathrm{KL}(\hat{\rho}\|\pi) + \ln\frac{1}{\delta}\right] + \frac{1}{2}(b-a)^2 .$$

Similarly, with $\lambda := \sqrt{n}$, Equation (11) becomes Equation (14) . $\qquad\square$

## A.4 Study of the Squared Loss

We consider a regression problem where $\mathcal{X}\times\mathcal{Y} \subset \mathbb{R}^d\times\mathbb{R}$, a family of linear predictors $f_{\mathbf{w}}(\mathbf{x}) = \mathbf{w}\cdot\mathbf{x}$, with $\mathbf{w}\in\mathbb{R}^d$, and a Gaussian prior $\mathcal{N}(\mathbf{0},\sigma_\pi^2\,\mathbf{I})$. Let us assume that the input examples are generated according to $\mathcal{N}(\mathbf{0},\sigma_{\mathbf{x}}^2\mathbf{I})$ and $\epsilon\sim\mathcal{N}(0,\sigma_\epsilon^2)$ is a Gaussian noise.

We study the squared loss $\ell_{\mathrm{sqr}}(\mathbf{w},\mathbf{x},y) = (\mathbf{w}\cdot\mathbf{x} - y)^2$ such that:

- $\mathbf{w}\sim\mathcal{N}(\mathbf{0},\sigma_\pi^2\,\mathbf{I})$ is given by the prior $\pi$,
- $\mathbf{x}\sim\mathcal{N}(\mathbf{0},\sigma_{\mathbf{x}}^2\mathbf{I})$ (and $\mathbf{x}\in\mathbb{R}^d$),
- $y = \mathbf{w}^*\cdot\mathbf{x} + \epsilon$, where $\epsilon\sim\mathcal{N}(0,\sigma_\epsilon^2)$, corresponds to the labeling function. Thus $y|\mathbf{x}\sim\mathcal{N}(\mathbf{x}\cdot\mathbf{w}^*,\sigma_\epsilon^2)$.

Let us consider the random variable $v = \left[\mathbf{E}_{\mathbf{x}}\mathbf{E}_{y|\mathbf{x}}\ell_{\mathrm{sqr}}(\mathbf{w},\mathbf{x},y)\right] - \ell_{\mathrm{sqr}}(\mathbf{w},\mathbf{x},y)$. To show that $v$ is a sub-gamma random variable, we will find values of $c$ and $s$ such that the criterion of Equation (16) is fulfilled, *i.e.*,

$$\psi_v(\lambda) \;=\; \ln\mathbf{E}\,e^{\lambda v} \;\leq\; \tfrac{\lambda^2 s^2}{2(1-c\lambda)}, \qquad \forall\lambda\in(0,\tfrac{1}{c}) .$$

We have,

$$\psi_v(\lambda) \;=\; \ln\mathbf{E}_{\mathbf{x}}\mathbf{E}_{y|\mathbf{x}}\mathbf{E}_{\mathbf{w}}\exp\left(\lambda\left[\mathbf{E}_{\mathbf{x}}\mathbf{E}_{y|\mathbf{x}}(y-\mathbf{w}\cdot\mathbf{x})^2\right] - \lambda(y-\mathbf{w}\cdot\mathbf{x})^2\right)$$

$$\leq \ln\mathbf{E}_{\mathbf{w}}\exp\left(\lambda\mathbf{E}_{\mathbf{x}}\mathbf{E}_{y|\mathbf{x}}(y-\mathbf{w}\cdot\mathbf{x})^2\right)$$

$$= \ln\mathbf{E}_{\mathbf{w}}\exp\left(\lambda\mathbf{E}_{\mathbf{x}}[\mathbf{x}\cdot(\mathbf{w}^*-\mathbf{w})]^2 + \lambda\sigma_\epsilon^2\right)$$

$$= \ln\mathbf{E}_{\mathbf{w}}\exp\left(\lambda\sigma_{\mathbf{x}}^2\|\mathbf{w}^*-\mathbf{w}\|^2 + \lambda\sigma_\epsilon^2\right)$$

$$= \ln\frac{1}{(1-2\lambda\sigma_{\mathbf{x}}^2\sigma_\pi^2)^{\frac{d}{2}}}\exp\left(\frac{\lambda\sigma_{\mathbf{x}}^2\|\mathbf{w}^*\|^2}{1-2\lambda\sigma_{\mathbf{x}}^2\sigma_\pi^2} + \lambda\sigma_\epsilon^2\right)$$

$$= -\frac{d}{2}\ln(1-2\lambda\sigma_{\mathbf{x}}^2\sigma_\pi^2) + \frac{\lambda\sigma_{\mathbf{x}}^2\|\mathbf{w}^*\|^2}{1-2\lambda\sigma_{\mathbf{x}}^2\sigma_\pi^2} + \lambda\sigma_\epsilon^2$$

$$\leq \frac{\lambda\sigma_{\mathbf{x}}^2\sigma_\pi^2 d}{1-2\lambda\sigma_{\mathbf{x}}^2\sigma_\pi^2} + \frac{\lambda\sigma_{\mathbf{x}}^2\|\mathbf{w}^*\|^2}{1-2\lambda\sigma_{\mathbf{x}}^2\sigma_\pi^2} + \lambda\sigma_\epsilon^2$$

$$= \frac{\lambda(\sigma_\pi^2\sigma_{\mathbf{x}}^2 d + \sigma_{\mathbf{x}}^2\|\mathbf{w}^*\|^2 + (1-2\lambda\sigma_{\mathbf{x}}^2\sigma_\pi^2)\sigma_\epsilon^2)}{1-2\lambda\sigma_{\mathbf{x}}^2\sigma_\pi^2}$$

$$= \frac{\lambda^2 s^2}{2(1-\lambda c)} ,$$

with $s^2 = \dfrac{2}{\lambda}\left[\sigma_{\mathbf{x}}^2(\sigma_\pi^2 d + \|\mathbf{w}^*\|^2) + \sigma_\epsilon^2(1-\lambda c)\right]$ and $c = 2\sigma_{\mathbf{x}}^2\sigma_\pi^2$ .

Recall that Corollary 5 is obtained with $\lambda := 1$.

## A.5 Linear Regression : Detailed calculations

Recall that, from Equation (25), the Gibbs optimal posterior of the described model is given by

$$p(\mathbf{w}|X,Y,\sigma,\sigma_\pi) = \mathcal{N}(\mathbf{w}\,|\,\widehat{\mathbf{w}}, A^{-1}) ,$$

with $A := \frac{1}{\sigma^2}\mathbf{\Phi}^T\mathbf{\Phi} + \frac{1}{\sigma_\pi^2}\mathbf{I}$ ; $\widehat{\mathbf{w}} := \frac{1}{\sigma^2}A^{-1}\mathbf{\Phi}^T\mathbf{y}$ ; $\mathbf{\Phi}$ is a $n{\times}d$ matrix such that the $i^{th}$ line is $\boldsymbol{\phi}(x_i)$ ; $\mathbf{y} := [y_1, \ldots y_n]$ is the labels-vector. For the complete derivation leading to this posterior distribution, see Bishop [5, Section 3.3] or Rasmussen and Williams [47, Section 2.1.1].

**Marginal likelihood.** We decompose of the marginal likelihood into the PAC-Bayesian trade-off:

$$- \ln p(Y|X, \sigma, \sigma_\pi)$$
$$= \frac{1}{2\sigma^2}\|\mathbf{y} - \mathbf{\Phi}\widehat{\mathbf{w}}\|^2 + \frac{n}{2}\ln(2\pi\sigma^2) + \frac{1}{2\sigma_\pi^2}\|\widehat{\mathbf{w}}\|^2 + \frac{1}{2}\log|A| + d\ln\sigma_\pi \qquad (\dagger)$$
$$= \underbrace{n\widehat{\mathcal{L}}_{X,Y}^{\ell_{\mathrm{nll}}}(\widehat{\mathbf{w}}) + \frac{1}{2\sigma^2}\mathrm{tr}(\mathbf{\Phi}^T\mathbf{\Phi}A^{-1})}_{n\,\mathbf{E}_{\mathbf{w}\sim\hat\rho^*}\,\widehat{\mathcal{L}}_{X,Y}^{\ell_{\mathrm{nll}}}(\mathbf{w})} + \underbrace{\frac{1}{2\sigma_\pi^2}\mathrm{tr}(A^{-1}) - \frac{d}{2} + \frac{1}{2\sigma_\pi^2}\|\widehat{\mathbf{w}}\|^2 + \frac{1}{2}\log|A| + d\ln\sigma_\pi}_{\mathrm{KL}\left(\mathcal{N}(\widehat{\mathbf{w}},A^{-1})\,\|\,\mathcal{N}(\mathbf{0},\sigma_\pi^2\mathbf{I})\right)} \quad . \quad (\star)$$

Line $(\dagger)$ corresponds to the classic form of the negative log marginal likelihood in a Bayesian linear regression context (see Bishop [5, Equation 3.86]).

Line $(\star)$ introduces three terms that cancel out : $\frac{1}{2\sigma^2}\mathrm{tr}\left(\mathbf{\Phi}^T\mathbf{\Phi}A^{-1}\right) + \frac{1}{2\sigma_\pi^2}\mathrm{tr}\left(A^{-1}\right) - \frac{1}{2}d = 0$ .
The latter equality follows from the trace operator properties and the definition of matrix $A$:

$$\frac{1}{2\sigma^2}\mathrm{tr}\left(\mathbf{\Phi}^T\mathbf{\Phi}A^{-1}\right) + \frac{1}{2\sigma_\pi^2}\mathrm{tr}\left(A^{-1}\right) = \mathrm{tr}\left(\frac{1}{2\sigma^2}\mathbf{\Phi}^T\mathbf{\Phi}A^{-1} + \frac{1}{2\sigma_\pi^2}A^{-1}\right)$$
$$= \mathrm{tr}\left(\frac{1}{2}A^{-1}(\frac{1}{\sigma^2}\mathbf{\Phi}^T\mathbf{\Phi} + \frac{1}{\sigma_\pi^2}\mathbf{I})\right)$$
$$= \mathrm{tr}\left(\frac{1}{2}A^{-1}A\right)$$
$$= \frac{1}{2}d \,.$$

We show below that the expected loss $\mathbf{E}_{\mathbf{w}\sim\hat\rho^*}\,\widehat{\mathcal{L}}_{X,Y}^{\ell_{\mathrm{nll}}}(\mathbf{w})$ corresponds to the left part of Line $(\star)$. Note that a proof of equality $\mathbf{E}_{\mathbf{w}\sim\hat\rho}\,\mathbf{w}^T\mathbf{\Phi}^T\mathbf{\Phi}\mathbf{w} = \mathrm{tr}\left(\mathbf{\Phi}^T\mathbf{\Phi}A^{-1}\right) + \widehat{\mathbf{w}}^T\mathbf{\Phi}^T\mathbf{\Phi}\widehat{\mathbf{w}}$ (Line ♣ below), known as the "expectation of the quadratic form", can be found in Seber and Lee [49, Theorem 1.5].

$$n\,\mathbf{E}_{\mathbf{w}\sim\hat\rho}\,\widehat{\mathcal{L}}_{X,Y}^{\ell_{\mathrm{nll}}}(\mathbf{w}) = \mathbf{E}_{\mathbf{w}\sim\hat\rho}\sum_{i=1}^{n} - \ln p(y_i|x_i, \mathbf{w})$$
$$= \mathbf{E}_{\mathbf{w}\sim\hat\rho}\left(\frac{n}{2}\ln(2\pi\sigma^2) + \frac{1}{2\sigma^2}\sum_{i=1}^{n}(y_i - \mathbf{w}\cdot\boldsymbol{\phi}(\mathbf{x}_i))^2\right)$$
$$= \frac{n}{2}\ln(2\pi\sigma^2) + \frac{1}{2\sigma^2}\mathbf{E}_{\mathbf{w}\sim\hat\rho}\|\mathbf{y} - \mathbf{\Phi}\mathbf{w}\|^2$$
$$= \frac{n}{2}\ln(2\pi\sigma^2) + \frac{1}{2\sigma^2}\mathbf{E}_{\mathbf{w}\sim\hat\rho}\left(\|\mathbf{y}\|^2 - 2\mathbf{y}\mathbf{\Phi}\mathbf{w} + \mathbf{w}^T\mathbf{\Phi}^T\mathbf{\Phi}\mathbf{w}\right)$$
$$= \frac{n}{2}\ln(2\pi\sigma^2) + \frac{1}{2\sigma^2}\left(\|\mathbf{y}\|^2 - 2\mathbf{y}\mathbf{\Phi}\widehat{\mathbf{w}} + \mathbf{E}_{\mathbf{w}\sim\hat\rho}\mathbf{w}^T\mathbf{\Phi}^T\mathbf{\Phi}\mathbf{w}\right)$$
$$= \frac{n}{2}\ln(2\pi\sigma^2) + \frac{1}{2\sigma^2}\left(\|\mathbf{y}\|^2 - 2\mathbf{y}\mathbf{\Phi}\widehat{\mathbf{w}} + \mathrm{tr}\left(\mathbf{\Phi}^T\mathbf{\Phi}A^{-1}\right) + \widehat{\mathbf{w}}^T\mathbf{\Phi}^T\mathbf{\Phi}\widehat{\mathbf{w}}\right) \quad (\clubsuit)$$
$$= \frac{n}{2}\ln(2\pi\sigma^2) + \frac{1}{2\sigma^2}\|\mathbf{y} - \mathbf{\Phi}\widehat{\mathbf{w}}\|^2 + \frac{1}{2\sigma^2}\mathrm{tr}\left(\mathbf{\Phi}^T\mathbf{\Phi}A^{-1}\right)$$
$$= n\widehat{\mathcal{L}}_{X,Y}^{\ell_{\mathrm{nll}}}(\widehat{\mathbf{w}}) + \frac{1}{2\sigma^2}\mathrm{tr}\left(\mathbf{\Phi}^T\mathbf{\Phi}A^{-1}\right) \,.$$

Finally, the right part of Line $(\star)$ is equal to the Kullback-Leibler divergence between the two multivariate normal distributions $\mathcal{N}(\widehat{\mathbf{w}}, A^{-1})$ and $\mathcal{N}(\mathbf{0}, \sigma_\pi^2\mathbf{I})$ :

$$\mathrm{KL}\left(\mathcal{N}(\widehat{\mathbf{w}}, A^{-1})\,\|\,\mathcal{N}(\mathbf{0}, \sigma_\pi^2\mathbf{I})\right) = \frac{1}{2}\left(\mathrm{tr}\left((\sigma_\pi^2\mathbf{I})^{-1}A^{-1}\right) + \frac{1}{\sigma_\pi^2}\|\widehat{\mathbf{w}}\|^2 - d + \log\frac{|\sigma_\pi^2\mathbf{I}|}{|A|}\right)$$
$$= \frac{1}{2}\left(\frac{1}{\sigma_\pi^2}\mathrm{tr}\left(A^{-1}\right) + \frac{1}{\sigma_\pi^2}\|\widehat{\mathbf{w}}\|^2 - d + \log|A| + d\ln\sigma_\pi^2\right) \,.$$

## A.6    Linear Regression: PAC-Bayesian sub-gamma bound coefficients

We follow then exact same steps as in Section A.4, except that we replace the random variable $v$ (giving the squared loss value) by a random variable $v'$ giving the value of the loss

$$\ell_{\text{nll}}(\langle\, \mathbf{w}, \sigma\,\rangle, \mathbf{x}, y) = \tfrac{1}{2}\ln(2\pi\sigma^2) + \tfrac{1}{2\sigma^2}(y - \mathbf{w}\cdot\mathbf{x})^2\,,$$

where $\mathbf{w}$, $\mathbf{x}$ and $y$ are generated as described in Section A.4. We aim to find the values of $c$ and $s$ such that the criterion of Equation (16) is fulfilled, *i.e.*,

$$\psi_{v'}(\lambda) \;=\; \ln\mathbf{E}\,e^{\lambda v'} \;\leq\; \tfrac{\lambda^2 s^2}{2(1-c\lambda)}\,, \qquad \forall \lambda \in \left(0, \tfrac{1}{c}\right).$$

We obtain

$$
\begin{aligned}
\psi_{v'}(\lambda) \;&=\; \ln\mathbf{E}_{\mathbf{x}}\,\mathbf{E}_{y|\mathbf{x}}\,\mathbf{E}_{\mathbf{w}}\exp\left(\tfrac{\lambda}{2\sigma^2}\big[\,\mathbf{E}_{\mathbf{x}}\,\mathbf{E}_{y|\mathbf{x}}(y-\mathbf{w}\cdot\mathbf{x})^2\big] - \lambda(y-\mathbf{w}\cdot\mathbf{x})^2\right) \\[4pt]
&\leq\; \ln\mathbf{E}_{\mathbf{w}}\exp\left(\tfrac{\lambda}{2\sigma^2}\,\mathbf{E}_{\mathbf{x}}\,\mathbf{E}_{y|\mathbf{x}}(y-\mathbf{w}\cdot\mathbf{x})^2\right) \\
&\;\;\vdots \\
&=\; \frac{\tfrac{\lambda}{2\sigma^2}\big(\sigma_\pi^2\sigma_{\mathbf{x}}^2 d + \sigma_{\mathbf{x}}^2\|\mathbf{w}^*\|^2 + (1 - 2\tfrac{\lambda}{2\sigma^2}\sigma_{\mathbf{x}}^2\sigma_\pi^2)\sigma_\epsilon^2\big)}{1 - 2\tfrac{\lambda}{2\sigma^2}\sigma_{\mathbf{x}}^2\sigma_\pi^2} \\[4pt]
&=\; \frac{\lambda^2 s^2}{2(1-\lambda c)}\,,
\end{aligned}
\tag{27}
$$

with

$$
\begin{aligned}
c \;&=\; \tfrac{1}{2\sigma^2}\Big[2\sigma_{\mathbf{x}}^2\sigma_\pi^2\Big] \;=\; \tfrac{1}{\sigma^2}(\sigma_{\mathbf{x}}^2\sigma_\pi^2)\,, \\[4pt]
s^2 \;&=\; \tfrac{1}{2\sigma^2}\Big[\tfrac{2}{\lambda}\big[\sigma_{\mathbf{x}}^2(\sigma_\pi^2 d + \|\mathbf{w}^*\|^2) + \sigma_\epsilon^2(1-\lambda c)\big]\Big] \;=\; \tfrac{1}{\lambda\sigma^2}\Big[\sigma_{\mathbf{x}}^2(\sigma_\pi^2 d + \|\mathbf{w}^*\|^2) + \sigma_\epsilon^2(1-\lambda c)\Big]
\end{aligned}
$$

## Supplementary Material References

[37] Luc Bégin, Pascal Germain, François Laviolette, and Jean-Francis Roy. PAC-Bayesian bounds based on the Rényi divergence. In *AISTATS*, pages 435–444, 2016.

[38] Yoav Freund and Robert E Schapire. A decision-theoretic generalization of on-line learning and an application to boosting. *Journal of Computer and System Sciences*, 55(1):119–139, 1997.

[39] Peter Grünwald and John Langford. Suboptimal behavior of Bayes and MDL in classification under misspecification. *Machine Learning*, 66(2-3):119–149, 2007.

[40] Peter Grünwald and Thijs van Ommen. Inconsistency of Bayesian Inference for Misspecified Linear Models, and a Proposal for Repairing It. *CoRR*, abs/1412.3730, 2014.

[41] Peter D. Grünwald. *The Minimum Description Length Principle*. The MIT Press, 2007. ISBN 0262072815.

[42] Sham M. Kakade and Andrew Y. Ng. Online bounds for Bayesian algorithms. In *NIPS*, pages 641–648, 2004.

[43] Simon Lacoste-Julien, Ferenc Huszar, and Zoubin Ghahramani. Approximate inference for the loss-calibrated Bayesian. In *AISTATS*, pages 416–424, 2011.

[44] John Langford and Matthias Seeger. Bounds for averaging classifiers. Technical report, Carnegie Mellon, Departement of Computer Science, 2001.

[45] Ron Meir and Tong Zhang. Generalization error bounds for Bayesian mixture algorithms. *Journal of Machine Learning Research*, 4:839–860, 2003.

[46] Andrew Y. Ng and Michael I. Jordan. On discriminative vs. generative classifiers: A comparison of logistic regression and naive Bayes. In *NIPS*, pages 841–848. MIT Press, 2001.

[47] Carl Rasmussen and Chris Williams. *Gaussian Processes for Machine Learning*. MIT Press, 2006.

[48] Judith Rousseau. On the frequentist properties of Bayesian nonparametric methods. *Annual Review of Statistics and Its Application*, 3:211–231, 2016.

[49] George A. F. Seber and Alan J. Lee. *Linear regression analysis*. John Wiley & Sons, 2012.