[Reviews · NeurIPS 2016]

Reviewer 1

Summary

Whereas most PAC-Bayesian results are not related to the Bayesian approach, the present article exhibits a connection: it finds a setting where minimizing the PAC Bayesian bound boils down to minimizing the marginal likelihood. The other contribution of the paper consists in improving PAC-Bayesian generalization bounds in the context of regression, where the loss functions considered are unbounded sub-Gamma functions. Use of these results for model selection purpose is also considered.

Qualitative Assessment

The paper is written with clarity and rigor. Its contribution is definitely novel and provides new perspectives on PAC Bayesian bound analysis. The sole "reproach" I could do is that the interest of considering sub-Gamma losses is not sufficiently motivated, which weakens the significance of the results to all appearances.

Confidence in this Review

3-Expert (read the paper in detail, know the area, quite certain of my opinion)


Reviewer 2

Summary

The paper presents (long desired) links between PAC-Bayesian theory and Bayesian inference, based on treating the negative log-likelihood as a loss function and the resultant bounds show a trade-off between the KL-divergence between the prior and posterior and the NLL. Technically this involves extending the normal analysis that only applies to bounded losses to the real line. Secondly, it is shown that Bayesian model selection through the model evidence can be linked to minimizing the PAC-Bayes bound. They show how this is applied to the special case of Bayesian linear regression.

Qualitative Assessment

The paper is well written and theoretically strong. It's been conjectured in the past that there should be links between PAC-Bayes theory and Bayesian inference, but to my knowledge this is the first theoretically complete demonstration of such links. Some comments: - In eq(8) (and above) the notion of a prior with bounded likelihood is introduced. Am I right in thinking that this is a data-dependent prior, since it can only be known if the likelihood will be bounded for a given prior after observing the data? If this is not the case can you explain how such a prior is possible? - I note that Theorem 3 is from unpublished work, although the proof is given as A.1. However I'm not sure the proof in A.1 is complete: Firstly \zeta_\pi and \Phi_{\ell, \pi, D} are not defined. Secondly, what are the requirements on X' and Y'? - in the proof of 14/15: -- where does the first inequality come from? -- it would be nice to state that you used the Chi-squared MGF during the proof, as it took me a while to work this out -- I couldn't quite work out why the equality was only valid for \lambda < \frac{1}{c}. Again I think this relates to the chi-squared MGF? Minor comments: - I'm not sure it helps to have so many references in the footnotes on p1. Perhaps if a numbered citation format was used rather than author-year, they could be put into the text - L22 do you mean implicitly rather than explicitly? - L95 Theorems 1 -> Theorem 1 - L185 mode -> model - L203 bounds -> bound - L206 is this L now the number of hyperparameters rather than the number of models? - L229 can be express by -> can be expressed as - L247 to a Gaussian -> by a Gaussian - Figure 1 panel (b) - legend is hard to read

Confidence in this Review

3-Expert (read the paper in detail, know the area, quite certain of my opinion)


Reviewer 3

Summary

The authors link the Bayesian and the PAC-Bayesian approaches by sufficient math derivation. They prove that for the negative log-likelihood loss function, minimizing PAC-Bayesian generalization bounds is equal to maximizing the Bayesian marginal likelihood. However, the paper is not easy to follow.

Qualitative Assessment

Possible improvements: 1: It is unclear how the related or previous work relate to and inform the current work. 2: The authors give many corollaries, but the experiment content is slightly inadequate, maybe some real data sets can be used to evaluate or validate. 3: In 265th line, 'minimizes' should be 'maximizes'.

Confidence in this Review

1-Less confident (might not have understood significant parts)


Reviewer 4

Summary

This paper compares PAC-Bayesian and Bayesian points of view. For the negative log-likelihood loss, the authors show that minimizing the PAC-Bayesian bound is equivalent to maximizing the bayesian marginal log likelihood. By this way, they extend PAC-Bayesian theorem to regression problem with unbounded loss.

Qualitative Assessment

This paper gives a new point of view using well-known ideas. The new results of Section 4 heavily rely on a paper by Alquier et al., leading to the analysis of model selection. In a Bayesian linear regression framework, the equality between the negative marginal log likelihood and the PAC-Bayesian bound using negative log-likelihood loss is also established. The numerical experiments are done on a toy example of the literature and show that both considered methods behave well for model selection. The new bound, derived in Section 4, is tighter than the one proposed in the literature. Minor issues: - l 167: In the paragraph Regression versus classification, I think it should be R(B)=0 instead of R(G)=0. - l 185: typo: each model - l 257: typo: than our sub-Gaussian result.

Confidence in this Review

3-Expert (read the paper in detail, know the area, quite certain of my opinion)


Reviewer 5

Summary

The paper provides a valuable bridge between Bayesian and PAC-Bayesian approaches and also provides useful bounds on the generalization error that appear to perform well in practice.

Qualitative Assessment

The work is very clearly written and the results are clearly stated and easy to use. These results will be of great value for my work and I suspect that many others will find them as useful as I do. For these reasons, I believe that this work should be accepted.

Confidence in this Review

2-Confident (read it all; understood it all reasonably well)